# Behavioral, Electrophysiological, and Toxicological Responses of *Plutella xylostella* to Extracts from *Angelica pubescens*

**DOI:** 10.3390/insects14070613

**Published:** 2023-07-06

**Authors:** Ruirui Zheng, Jinyu Zhao, Li Ma, Xingtao Qie, Xizhong Yan, Chi Hao

**Affiliations:** College of Plant Protection, Shanxi Agricultural University, Jinzhong 030800, China; zhengruirui2019@163.com (R.Z.); zhaojinyu312@163.com (J.Z.); mali890310@126.com (L.M.); qxt123@nwafu.edu.cn (X.Q.)

**Keywords:** *Plutella xylostella*, *Angelica pubescens*, extract/compounds, toxicity, repellent effect

## Abstract

**Simple Summary:**

*Plutella xylostella* is one of the most destructive insect pests affecting cruciferous vegetables. The use of various plant extracts for controlling this pest is increasing in popularity, as they are multi-bioactive, biodegradable, and ecologically safe. In this study, we evaluated the insecticidal oviposition deterrence and repellent activities of *Angelica pubescens* extract and its compounds against *P. xylostella*. The *A. pubescens* extract, caryophyllene oxide, and 3,4-dimethoxycinnamic acid, have been proven to have insecticidal and oviposition deterrent activities against *P. xylostella*. The *A. pubescens* extract and caryophyllene oxide also mediate *P. xylostella* repellence. Overall, our results highlighted the potential of *A. pubescens* extract, caryophyllene oxide, and 3,4-dimethoxycinnamic acid in controlling *P. xylostella* and in the development of effective eco−friendly “inhibit and kill” insecticide formulations.

**Abstract:**

*Plutella xylostella* L. is a destructive pest affecting cruciferous vegetables, causing massive economic losses worldwide. Plant−based insecticides are considered promising insect control agents. The *Angelica pubescens* extract inhibited female oviposition, with an oviposition deterrence index (ODI) of 61.65% at 12.5 mg/mL. We aimed to identify the bioactive compounds in *A. pubescens* extract. The compounds from *A. pubescens* extract were analyzed using LC−MS techniques. The toxicity and behavioral responses of larvae and adults of *P. xylostella* to ten compounds were investigated. We found that the caryophyllene oxide and 3,4-dimethoxycinnamic acid inhibited female oviposition; the ODIs were 98.31% and 97.59% at 1.25 mg/mL, respectively. The *A. pubescens* extract, caryophyllene oxide, and 3,4-dimethoxycinnamic acid caused larval mortality, with LC_50_ values of 21.31, 4.56, and 5.52 mg/mL, respectively. The EAG response of females was higher than that of males under *A. pubescens* extract conditions, while the EAG response of males was higher than that of females in caryophyllene oxide and 3,4-dimethoxycinnamic acid conditions. The *A. pubescens* extract and caryophyllene oxide showed repellent activity against both female and male adults, while the 3,4-dimethoxycinnamic acid did not elicit any notable behavioral responses from *P. xylostella* adults. *A. pubescens* extract and caryophyllene oxide are potential insecticides, oviposition deterrents, and behavioral regulators against *P. xylostella*, and they could be potential candidates for the development of biological insecticides to control *P. xylostella*.

## 1. Introduction

*Plutella xylostella* (Lepidoptera: Plutellidae) is one of the most damaging insect pests of cruciferous vegetables in global agricultural ecology [1]. *P. xylostella* adults have high fecundity and create a large larval population, and third and fourth-instar *P. xylostella* larvae have high feeding rates, causing direct damage to cruciferous vegetables and an estimated annual global economic loss of USD 4–5 billion [2]. Targeting the adult and larval stages of *P. xylostella* (i.e., using insect behavior-regulating stimuli to manipulate population structures and kill larvae) has been shown to be effective. Over the past few decades, due to excessive dependence on chemical pesticides to control *P. xylostella*, *P. xylostella* has developed a resistance to the traditional insecticides used to control it [3]. In addition, synthetic pesticides contain high levels of toxic residues, exert adverse effects on non−target organisms, and threaten human and environmental safety [4]. Thus, the development of new and effective “green” repellents and insecticides with different mechanisms of action could circumvent the current challenges in effectively controlling *P. xylostella* [5]. One benign alternative to synthetic insecticides is plant extracts and their derived compounds. Crude plant extracts could control *P. xylostella* in various ways, including via insecticidal [6] and repellent effects, growth and oviposition inhibition [7], and reducing their feeding [8]. Beyond their insecticidal properties, plant extracts have several advantages over traditional insecticides. They are generally considered safe for human and animal use, are biodegradable, and pose low toxicity risks for non-target organisms; these factors make them more environmentally acceptable than several synthetic insecticides.

Most moths rely on compounds from host plants to locate food resources and suitable oviposition sites. Compounds from non-host plants can cover and mask these host-plant-derived compounds, influencing orientation behaviors. Based on the literature, 42 different Apiaceae plant extracts have previously been demonstrated to possess biological insecticide activities, including *Foeniculum vulgare* essential oil’s acute toxicity to *Myzus persicae* [9], *Trachyspermum ammi* essential oil’s repellence activity to *Riptortus clavatus* [10], and *Angelica archangelica* extract, which has insecticidal, antifeedant, and growth inhibition activities against *Spodoptera littoralis* larvae [11]. The chemical compounds from Apiaceae plants, including sesquitterpenic lactones, furanocoumarins, monoterpene coumarins, polyacetylenes, volatiles pheylpropenes, and phethalides, have also been shown to possess a range of biological activities, including insecticidal, antifungal, and antioxidant properties [12,13]. Thus, Apiaceae plant extracts have considerable potential for application as biopesticides in the management of pests. *A. pubescens* belongs to the Apiaceae family, and its root is used in traditional Chinese medicine. Clinical applications have focused on its pharmacological activities, such as its analgesic, sedative, and hypnotic effects [14]. In previous studies, *A. pubescens* has been shown to have deworming [15] and insecticidal properties [16]. However, the oviposition deterrence and insecticidal activities of *A. pubescens* against *P. xylostella* and other Lepidopteran insects have not been reported.

The application of plant-extract-derived bioactive compounds as alternatives to synthetic insecticides has received considerable attention. *A. pubescens* extract contains coumarins, polyene−alkynes, phenolic acids, steroids, nucleoside elements, and other chemical compounds [15]. The main volatile compounds from the *A. pubescens* extract comprise polyene−alkynes and phenolic acids. For instance, the polyene−alkyne compounds alantolactone, isoalantolactone [17], and 1−hexene [18], and the phenolic acid compound caffeic acid, showed growth inhibiting and repellent effects against *Spodoptera litura* larvae and adults [19]. Thus, polyene−alkyne and phenolic acid compounds have potential in controlling *P. xylostella*. We studied the biological activities of *A. pubescens* extract and its constituents against *P. xylostella*, and conducted a preliminary study on its functional mechanisms. 

In this study, for the first time, a hydroalcoholic extract was prepared from *A. pubescens*, a novel plant preparation with oviposition deterrence and insecticidal activities against *P. xylostella*. We used LC−MS techniques to identify and quantify the constituents of this *A. pubescens* extract. We aimed to test the following: (1) whether the constituents of *A. pubescens* extract have oviposition deterrence activity against *P. xylostella* adults and insecticidal activity against *P. xylostella* larvae (using the dual-choice oviposition deterrent bioassay and leaf dipping method); (2) the olfactory responses of *P. xylostella* to these bioactive compounds, investigated by electrophysiological tests, to assess the sensitivity of female and male antennae to the test compounds and a Y-tube olfactometer bioassay to evaluate the insects’ behavioral response of the same compounds. Our findings regarding hypothesis (2) will help to further elucidate the relationship between the oviposition deterrent activity of these compounds and olfactory behavior of *P. xylostella*. This study provides preliminary data for elucidating the mechanisms of action of *A. pubescens* extract and its constituents to control *P. xylostella* adults and larvae.

## 2. Materials and Methods

### 2.1. Insects 

*Plutella xylostella* larvae were collected from the vegetable fields of Shanxi Agricultural University Experimental Station in Taigu County, Jinzhong City, Shanxi Province, China (37°25′22″ N, 112°34′15″ E). Multi-generation larvae were successively reared in the laboratory of Shanxi Agricultural University, following the rearing protocol previously described by Yan et al. [20] and Guo et al. [21]. Larvae were fed on fresh cabbage leaves (*Brassica oleracea* (Gai Liang Qing Fen)). The moths were reared in a 30 × 40 × 60 cm cage (consisting of a plastic tube frame and fitted with insect-proof screens) and fed with a 15% honey solution. Cabbage leaves were placed in the cage to facilitate egg-laying by the moths. All pupae were placed in a 10 cm glass tube with a cotton plug to ensure access to fresh air. The insect house was maintained at a temperature of 25 ± 2 °C, relative humidity of 60 ± 10%, and photoperiod of 16:8 h (L:D). Experiments were performed on *P. xylostella* adults (24–48 h post-emergence) and on neonate uniform third instar larvae, which were starved for 4 h before experiments. The F10 generation was used for all bioassays.

### 2.2. Chemicals 

The chemical compounds used for the bioassays were purchased from Aladdin (Shanghai, China) and Macklin (Shanghai, China) (Appendix A). LC−MS grade methanol, acetonitrile, and formic acid were purchased from Merck KGaA (Germany) and Xiya Reagent (Shandong, China). Matrine (Hei Ye Ming) was purchased from Zhenge Biotechnology (Guangdong, China). 20% ethanol, 0.5% Triton−X100, and 0.5% DMSO (TDE solution) mixture was used as a solvent, and the compound test solutions and matrine were configured as 10, 5, 2.5, 1.25, and 0.63 mg/mL.

### 2.3. Extraction and LC−MS Analysis

Commercial *A. pubescens* root material was purchased from Renhe Pharmaceutical Co., Ltd. (Jiangxi, China) and ground to a powder using a grinding machine. As described by the extraction method of Domenico et al. [22], with slight modifications. Then, 300 g of the powder was dissolved in 600 mL of solvent (25% water, 25% 2-propanol, and 50% ethanol). This mixture was agitated overnight at room temperature (30 °C, September, Taigu, China) before sonication treatment for 30 min at 40 °C. The supernatant was filtered through filter paper with a vacuum suction filter. The sonication and filtration processes were repeated five times. The solvents were combined and evaporated under reduced pressure at 50 °C to obtain 100 mL extract solvents, which were stored in a refrigerator at 4 °C until further use.

*A. pubescens* extract was diluted in TDE solution to obtain 200, 100, 50, 25, and 12.5 mg/mL test solutions. Subsequently, the extract was vigorously mixed using a vortex machine for 5 min to dissolve it thoroughly. 

LC−MS of the *A. pubescens* extract was performed by the Sci−Tech Innovation Quality Testing Company (Qingdao, China). The LC−MS analysis method is outlined in the Appendix A.

### 2.4. Dual-Choice Oviposition Assay 

The oviposition deterrence assay method was performed as described by Yang et al. [23]. The assay was carried out using 5 sets of moths, 5 females and 15 males, which were introduced into a transparent plastic cylindrical lidded container (20 cm diameter × 8 cm height, with at least 40 pinholes for ventilation). The *A. pubescens* extract (200, 100, 50, 25, and 12.5 mg/mL) and its derived compounds of caryophyllene oxide and 3,4-dimethoxycinnamic acid (10, 5, 2.5, 1.25, and 0.63 mg/mL) were used as test solutions. The TDE solution was used as a negative control. The 10 mg/mL matrine solution was used as the positive control. A cabbage leaf disc (6 cm in diameter) was placed into the test solution (5 mL) for 60 s, and the solvent on the leaf disc was allowed to dry naturally via evaporation. The leaf disc was placed at a diagonal end position in a clear transparent plastic cylindrical container. We placed a small piece of absorbent cotton (soaked with 15% honey) in the center of the clear transparent plastic cylindrical container. Three replicates were prepared for each test solution. The number of eggs on the cabbage leaf disc was counted after 24 h of treatment. 

### 2.5. Insecticidal Assay 

The insecticidal assay was performed using the leaf dip method, as reported by Ratnaweera et al. [24]. The *A. pubescens* extracts (200, 100, 50, 25, and 12.5 mg/mL) and derived compounds of caryophyllene oxide and 3,4-dimethoxycinnamic acid (10, 5, 2.5, 1.25, and 0.63 mg/mL) were used as test solutions. A cabbage leaf disc (6 cm in diameter) was placed into the test solution (5 mL) for 60 s, the solvent was evaporated naturally, and the disc was put into a Petri dish padded with moistened filter paper. TDE solution was used as a negative control and 10 mg/mL matrine solution was maintained as a positive control. Twenty third-instar larvae were placed in the Petri dish. Five replicates were performed for each test solution. The numbers of dead larvae were counted at 24, 48, and 72 h of treatment. The larvae were considered dead when they demonstrated paralysis, tipping, and/or immobility when touched with the bristles of a fillet-type brush.

### 2.6. Electrophysiological (EAG) Assay

We used the EAG technique as descried by Song et al., with some modifications [4], to test the antennal response of *P. xylostella* adults to *A. pubescens* extract (200, 100, 50, 25, and 12.5 mg/mL) and its constituents caryophyllene oxide and 3,4-dimethoxycinnamic acid (10, 5, 2.5, 1.25, 0.63 mg/mL). Ice was used to reduce the moth’s mobility before excising the antenna at the base of the flagellum and cutting 2 mm from the antenna tip. The antenna was positioned between two sharpened electrode probes (reference and record electrodes) containing a conducting gel. The TDE solution was used as a control to measure the potential antenna response. We pipetted 20 µL of test solution onto a piece of 0.5 × 5 cm Whatman^®^ filter paper (the concentrations were 1.6, 0.8, 0.4, 0.2, and 0.1 g/cm^2^ and 0.08, 0.04, 0.02, 0.01, and 0.005 g/cm^2^) and inserted it into a Pasteur pipet (15 cm in length). The Pasteur pipet device was connected to the stimulation controller via a silicone tube. When the baseline was stable, each antenna was calibrated with the control solution, followed by the test solutions at five concentrations from low to high, then the control again. The duration of each stimulation was 0.5 s, with 30 s intervals between successive stimulations. For the extract and each constituent sample, new sets of six virgin females and six virgin males (with two replications of each stimulus) were tested.

### 2.7. Y-Tube Olfactometer Assay 

The behavioral response of *P. xylostella* adults to *A. pubescens* extract (200, 100, 50, 25, and 12.5 mg/mL) and its constituents caryophyllene oxide and 3,4-dimethoxycinnamic acid (10, 5, 2.5, 1.25, and 0.63 mg/mL) were tested using a Y-tube olfactometer (internal width of 24 mm × length of 20 cm (choice arm) and 29 mm × 20 cm (main arm) with a 30° angle between arms), as described by Song et al. [4]. For the treatment arm, 20 µL of test solution was applied to a piece of Whatman^®^ filter paper (5 cm in length and 5 mm in width). For the control arm, filter paper containing 20 µL 15% honey water was provided. The *P. xylostella* adults were placed at the main arm.

Using an air sampler, atmospheric air was allowed to flow through an activated charcoal cartridge. Purified air was pushed into the Y-tube olfactometer at a rate of 300 mL/min per pump. Each arm was ventilated for 3 min to fill it with test gas prior to the experiment. During the test, a moth was considered to have made a choice when it crossed the midpoint of one of the upwind choice arms and stayed there for more than 30 s. If a moth did not move after 5 min in the main arm, it was recorded as “no choice”. To avoid position effect bias, the test solvents were switched between arms when 10 insects had made their choice. Behavioral tests were performed from 14:00 to 18:00, under red light and room-temperature conditions (25 ± 1 °C). Olfactometers were cleaned at the end of each day with bleach and water, and left to dry overnight. In total, 50 virgin males and 50 virgin females were tested for each test solvent.

### 2.8. Statistical Analysis 

The oviposition deterrence index (ODI) of adults and the calibrated mortality rate of larvae were calculated using Abbott’s formula [25]. The ODI, calibrated mortality, and EAG response data were subjected to one-way ANOVA, followed by Tukey’s multiple comparison tests to determine significant differences (*p* < 0.05). The numbers of eggs in the treatment and control groups were subjected to independent-samples t-tests to determine significant differences. Probit analyses of larvae-calibrated mortality data were performed to determine the median lethal dose (LC_50_) and corresponding 95% confidential intervals.
$$Calibrated\;mortality=\frac{rate\;of\;mortality\;in\;the\;treatment-rate\;of\;mortality\;in\;the\;control}{1-rate\;of\;mortality\;in\;the\;control}$$
$$\mathrm{Oviposition}\;\mathrm{deterrence}\;\mathrm{index}=\frac{\mathrm{number}\;\mathrm{of}\;\mathrm{eggs}\;\mathrm{in}\;\mathrm{the}\;treatment-\mathrm{number}\;\mathrm{of}\;\mathrm{eggs}\;\mathrm{in}\;\mathrm{the}\;control}{\mathrm{number}\;\mathrm{of}\;\mathrm{eggs}\;\mathrm{in}\;\mathrm{the}\;treatment+\mathrm{number}\;\mathrm{of}\;\mathrm{eggs}\;\mathrm{in}\;\mathrm{the}\;control}$$

EAG response values were calculated as (b − a)/a × 100, where a is the average EAG amplitude value of the control stimulus and b is the average EAG amplitude value of treated stimulus. 

For the behavioral results, the proportion of *P. xylostella* which chose the treatment and control was calculated. Chi-squared tests were used to determined significant differences. Any moths that did not make a definitive choice were not used in the statistical analysis.

## 3. Results

### 3.1. Bioassay Activity and LC−MS Analysis for A. pubescens Extract 

In previous study that had tested 20 plant species, the *A. pubescens* extract demonstrated oviposition deterrent and insecticidal activities against *P. xylostella*, with an oviposition deterrence index of 96.88% at 100 mg/mL, and the larval mortality rate was 97.00% at 100 mg/mL (Table 1). The polyene−alkynes and phenolic acids were the main volatile constituents of *A. pubescens* extract. Thus, we selected ten compounds from the polyene−alkynes and phenolic acids, with contents of at least 0.01% in *A. pubescens* extract, for subsequent experiments. The retention time, composition content, and chemical structures are presented in Table 2 and Appendix A.

### 3.2. Oviposition Deterrent Activity of 10 Compounds against Female Adults of P. xylostella

The results showed that the 10 compounds (at a concentration of 10 mg/mL) significantly (*p* < 0.05) affected the oviposition behavior of female *P. xylostella* adults. Caryophyllene oxide, nootkatone, and 4-coumaric acid showed significant oviposition deterrent activity against *P. xylostella* females, with ODI values of 100%, 100%, and 90.67% at 10 mg/mL, respectively, which were 1.3–1.5 times higher than that of the positive control matrine. The ODI of 3,4-dimethoxycinnamic acid against *P. xylostella* females was 57.01% at 10 mg/mL, which was 0.8 times higher than that of the positive control matrine. The ODIs of other compounds against *P. xylostella* females did not exceed 50% at 10 mg/mL (Table 3).

### 3.3. Insecticidal Activity of 10 Compounds against P. xylostella Larvae 

The 72 h mortality results showed that the caryophyllene oxide and 3,4-dimethoxycinnamic acid (at 10 mg/mL) had a calibrated mortality above 75%, which was 0.75 times higher than that of the positive control matrine. In contrast, the mortality under other compounds did not exceed 55% (Table 3). 

### 3.4. Oviposition Deterrent and Insecticidal Activities of A. pubescens Extract, Caryophyllene Oxide, and 3,4-Dimethoxycinnamic Acid against P. xylostella

Based on the results of screening assays, *A. pubescens* extract, caryophyllene oxide, and 3,4-dimethoxycinnamic acid exhibited oviposition deterrent activity against *P. xylostella* female adults and insecticidal activity against *P. xylostella* larvae. Thus, these compounds were used in subsequent experimental studies.

The LC_50_ values of the *A. pubescens* extract, caryophyllene oxide, and 3,4-dimethoxycinnamic acid in the larval stages were 21.31, 4.56, and 5.52 mg/mL, respectively (Table 4). The cabbage leaves treated with 200 mg/mL *A. pubescens* extract had no eggs and the total number of eggs was reduced; as the concentration decreased, the number of eggs of *P. xylostella* increased. In addition, no eggs were found on the cabbage leaves treated with caryophyllene oxide, and the total numbers of eggs were reduced at concentrations of 10, 5, and 2.5 mg/mL. The 3,4-dimethoxycinnamic acid demonstrated oviposition deterrence activity against female *P. xylostella* at 1.25 mg/mL, which surpassed that at the higher concentrations (10 mg/mL, 5 mg/mL, and 2.5 mg/mL), but the total number of eggs was reduced at concentrations of 5, 2.5, and 1.25 mg/mL (Figure 1).

### 3.5. Differential EAG Responses to A. pubescens Extract and Individual Compounds

EAG responses were elicited by concentrations of 200, 100, 50, 25, and 12.5 mg/mL *A. pubescens* extract, concentrations of 10, 5, 2.5, 1.25, and 0.63 mg/mL caryophyllene oxide, and 3,4-dimethoxycinnamic acid treatment (Figure 2). In both sexes, the EAG responses were higher under different concentrations of caryophyllene oxide than these under 3,4-dimethoxycinnamic acid. In the *A. pubescens* extract treatment group, the EAG responses indicated that the female moths were slightly more affected than the male moths, with EAG values of 4.86 (female) and 1.13 mV (male) at 200 mg/mL, respectively. In the caryophyllene oxide treatment group, the EAG response of female adults was slightly lower than that of male adults at concentrations of 2.5, 1.25, and 0.63 mg/mL. The EAG responses of male adults treated with caryophyllene oxide at 0.63 mg/mL were higher than at the concentrations of 5, 2.5, and 1.25 mg/mL, which indicated that male receptors were more sensitive at lower concentrations. The EAG responses of *P. xylostella* treated with 3,4-dimethoxycinnamic acid showed that the female moths were less sensitive than the male moths, with EAG values of 1.00 (females) and 1.19 mV (males) at 10 mg/mL, respectively. The EAG responses of female moths treated with 3,4-dimethoxycinnamic acid did not differ significantly at 5, 2.5, 1.25, and 0.63 mg/mL, with values of 0.46, 0.59, 0.35, and 0.49 mV, respectively.

### 3.6. Differential Behavioral Effects of A. pubescens Extract and Individual Compounds 

The *A. pubescens* extract showed a significant repellent effect (*p* < 0.05) for both male and female adults of *P. xylostella* at a concentration of 25 mg/mL. There was a significant reduction in female *P. xylostella* who selected the *A. pubescens* extract arm (*p* < 0.05) at 100 mg/mL when compared with these which chose the control arm. Caryophyllene oxide at concentrations of 5 and 2.5 mg/mL showed a significant repellent effect for both sexes of *P. xylostella* (*p* < 0.05), and female *P. xylostella* preferred the control when the concentration was 0.63 mg/mL. Males also significantly preferred the control arm (*p* < 0.05) at 10 mg/mL. However, no significant differences in choice were observed for either sex when choosing between 3,4-dimethoxycinnamic acid at five concentrations or the control, except for the male selection control arm at 5 mg/mL (Figure 3).

## 4. Discussion

The indiscriminate use of synthetic insecticides has caused a series of adverse effects on the environment and consumers. Novel pesticides to control *P. xylostella* are urgently needed to combat these environmental safety challenges. Previous studies have shown that plant extracts and derived bioactive compounds have a variety of biological activities to control pests, and they provide an alternative to synthetic insecticides due to their biodegradable nature and safety for non-target organisms [7,26,27].

Solvent-based (water, 2-propanol, and ethanol) sonication extraction methods have commonly been used to extract compounds from Chinese medicinal plants. This method is attractive because it avoids the use of other potentially harmful solvents, making the process more eco-friendly and safer for use in the field.

In our study, *A. pubescens* extract exhibited oviposition deterrence activity against *P. xylostella* adults, with an ODI of 61.50% at 12.5 mg/mL, and insecticidal activity against *P. xylostella* larvae, with an LC_50_ value of 21.31 mg/mL. Compared with other extracts, the ethanol/distilled water extraction of *Trillium govanianum* exhibited insecticidal activity against larvae of *P. xylostella*, with mortality not exceeding 50% at 72 h and an LC_50_ value of 11.040 mg/mL at 96 h [28]. The *Lantana camara* extracts at 8% and 10% showed ovicidal and oviposition deterrence activities, with a lower insecticidal activity against *P. xylostella* larvae [29]. The *n*-hexane fraction of *Zanthoxylum armatum* extract showed insecticidal activity against larvae of *P. xylostella* with an LC_50_ value of 29.88 mg/mL, which was similar to what was achieved in the present study [30]. Thus, *A. pubescens* extract can be used as a potential oviposition deterrent and insecticidal agent to control *P. xylostella.*

Active compounds derived from plants or plant products have important bioactive functions and potential applications in agricultural pest management [4,27,31]. They often have a similar mode of action to synthetic insecticides [32], and may be able to provide precursor compounds for synthetic pesticides. Caryophyllene oxide is a common polyene−alkyne that has been identified as one of the main constituents of various extracts with biological activities. For example, it is one of the main constituents of *Cyperus rotundus* essential oil, which showed contact toxicity and repellent activity against *Callosobruchus maculatus*, *Trogoderma granarium*, and *Oryzaephilus surinamensis* [33]. It is also a major constituent of *Lantana camara* extract, which had ovicidal activity and an oviposition deterrent effect on *P. xylostella* [29]. In addition, Huang et al. [31] reported that *Artemisia lavandulaefolia* extract and its constituent caryophyllene oxide had strong contact toxicity and fumigant activity against larvae and adults of *P. xylostella*, with LD_50_ values of 20.71 mg/mL and 1.06 mg/mL, respectively. In our study, caryophyllene oxide exhibited oviposition deterrence and insecticidal activities against *P. xylostella* adults and larvae, with an ODI of 98.31% at 1.25 mg/mL and an LC_50_ value of 4.59 mg/mL. The above findings suggested that caryophyllene oxide has multiple modes of action against various insects, including *P. xylostella.* Yulia [34] summarized the biological activity of β-caryophyllene and its oxidized form; these compounds can easily penetrate cell membranes [35], do not cause allergic reactions upon contact, even in persons with dermatitis [36], increased the cytochrome P450 activity of mice [37], and exhibited in vivo pain-alleviating and anti-inflammatory properties [38]. Thus, caryophyllene oxide could be a candidate in the development of new insecticides. 

3,4-dimethoxycinnamic acid is a common phenolic acid, with medicinal value against tumor cells, methicillin-resistant *Staphylococcus aureus*, and *Plasmodium falciparum* [39]. In our study, 3,4-dimethoxycinnamic acid showed oviposition deterrence and insecticidal activities against *P. xylostella*, with an ODI of 97.59% at 1.25 mg/mL and an LC_50_ value of 5.52 mg/mL. 3,4-dimethoxycinnamic acid is functionally related to trans-cinnamic acid (NCBI−Pub chem), which showed larvicidal activity against *Aedes aegypti* [40] and nematicidal activities against *Meloidogyne incognita* and *Tylenchulus semipenetrans* [41]. In addition, cinnamic acid derivatives with a methylxanthine caffeine skeleton showed insecticidal potential against *Mythimna separata* [42]. Importantly, 3,4-dimethoxycinnamic acid is a key constituent of propolis, which is not cytotoxic to normal cell lines [39]. 

In addition, we observed that 3,4-dimethoxycinnamic acid showed oviposition deterrence activity against *P. xylostella*, with an ODI of 97.59% at 1.25 mg/mL. The biological effects decreased as the concentration increased, with an ODI of 57.01% at 10 mg/mL. Previous reports regarding the effects of these volatile compounds at different concentrations have reported varying results. For instance, (Z)-9-octadecenoic acid ethyl ester (OE) induced an oviposition preference and exhibited concentration-dependent bidirectional regulation. High concentrations of OE inhibited oviposition, while low doses promoted oviposition, with the Gr32a playing a role in oviposition stimulation [43]. Linalool (10 and 100 mg/L), and cis-3-hexen-1-ol (100 mg/L) exhibited repellent activities against *Cyrtorhinus lividipennis*. However, linalool (1 mg/L), and cis-3-hexen-1-ol (0.1 mg/L) appeared to attract *C. lividipennis* [44]. Additionally, the egg-laying preference of *Helicoverpa assulta* for heptanal disappeared when the concentration increased from 0.001 mol/L to 0.01 mol/L [45]. Nevertheless, our results showed that 3,4-dimethoxycinnamic acid has potential as an insecticide against *P. xylostella* at certain concentrations.

Numerous studies have shown that non-host plant extracts and derived compounds can cover or mask host plant compounds. This depends on the chemical communication involved in *P. xylostella* egg-laying and feeding. For instance, *Traiadica sebifera* extract and its fractions showed antifeedant activities against *P. xylostella* larvae [27]. *Ludwigia tomentosa* and *L. longifolia* extracts showed oviposition deterrence and insecticidal activities against *P. xylostella* [7]. Generally, insects utilize olfaction from their antenna to perceive and distinguish semiochemicals [46]. In our study, the EAG and behavioral responses of *P. xylostella* to *A. pubescens* extract and its compounds, caryophyllene oxide and 3,4-dimethoxycinnamic acid, were studied via EAG and Y-tube olfactometer experiments.

The EAG response is linked to the number of olfactory receptors in the insect’s antennae [47]. In this study, both female and male adults of *P. xylostella* showed positive EAG responses to *A. pubescens* extract, caryophyllene oxide, and 3,4-dimethoxycinnamic acid, with responses ranging from 0.4 to 5.0 mV. For comparison, the EAG responses of *P. xylostella* to α-farnesene and β-myrcene ranged from 0.4 to 1.2 mV and from 0.1 to 1 mV, respectively [48]. Overall, the *P. xylostella* olfactory sensilla of both sexes were able to perceive *A. pubescens* extract, caryophyllene oxide, and 3,4-dimethoxycinnamic acid, which indicates that *A. pubescens* extract and its compounds are relevant in both female and male ecology. The EAG responses indicated a higher neuronal response to caryophyllene oxide than 3,4-dimethoxycinnamic acid in *P. xylostella* adults, which was correlated with increased receptor stimulation, and may have affected their behavioral responses.

In our study, the EAG response of female adults was more sensitive than that of male adults to *A. pubescens* extract. However, the male moths had slightly higher sensitivity than female moths in EAG response to different concentrations of caryophyllene oxide and 3,4-dimethoxycinnamic acid. Previous research has shown that this sexually dimorphic EAG response is linked to the volatile semiochemical detection cues of different numbers of olfactory receptors in male and female antennae, due to the different roles played by the same compound in the ecology of the different sexes [49]. The female insects use plant volatile compounds to find suitable oviposition positions [50]; the male insects use plant volatile compounds to find environments where females may be present [51]. For instance, female adults of *Cotesia glomerata* perceived more compounds than the males when undergoing EAG response assessments to benzyl cyanide [52]. The females also showed a higher EAG sensitivity to allyl isothiocyanate than males [48]. Thus, we hypothesized that *A. pubescens* extract may influence the oviposition behavior of *P. xylostella* females. In addition, previous studies have found that females of *P. xylostella* showed lower EAG responses than males to α-farnesene, β-myrcene, octanal, nonanal, and methyl jasmonate (at concentrations of 0.1) [48], while (Z)-3-hexen-1-ol elicited higher EAG responses in males of *Crioceris duodecimpunctata* than females [53]. In our study, the EAG responses of male moths to caryophyllene oxide and 3,4-dimethoxycinnamic acid were stronger than the responses of female moths; these two compounds may play an important role in interfering with the recognition of intraspecific sex pheromones by male moths.

Compounds that can trigger EAG activity in insect antennae do not necessarily lead to behavioral responses [26]. Therefore, olfactory assays were required to determine the behavioral responses of *P. xylostella* to *A. pubescens* extract and its compounds. In the Y-tube olfactory assays, female and male *P. xylostella* adults preferred the control arm over the treated arm with *A. pubescens* extract and caryophyllene oxide, which exhibited repellent effects and led to different behaviors at different concentrations. Previous research found that green leaf volatiles at different concentrations induced variable behavioral effects in European corn borer larvae [54]. Terpinene-4-ol and camphor showed concentration-dependent repellent activities against *P. xylostella* females [4], similarly to our results. The repellent response of *P. xylostella* adults to *A. pubescens* extract and caryophyllene oxide was due to the ability of their antennae to perceive these odorous substances. Some previous studies have reported that *Nilaparvata lugens* specimens were repelled by caryophyllene oxide, and *Nlug*OBP8 was highly expressed in multiple sensilla of *N. lugens* antenna. Antennae of ds*Nlug*OBP-injected BPHs completely lost the ability to sense caryophyllene oxide [55,56]. Nguyen et al. reported that caryophyllene oxide was ranked highly as a mosquito odorant-binding protein inhibitor (with larvicidal activity) based on its docking scores, and likely docks similar to permethrin [57]. Thus, we hypothesized that caryophyllene oxide has repellent activity against *P. xylostella* and may affect the odorant-binding protein of *P. xylostella*. Relevant research has reported that repellents prevent both female and male moths from achieving adequate orientation/location on the surfaces of treated leaves, thereby reducing the mating rates and egg-laying numbers [58]. The *A. pubescens* extract at a concentration of 25 mg/mL, caused a repellent effect for both female and male adults of *P. xylostella*, without influencing the overall egg-laying ability of *P. xylostella* female adults, which indicated that *A. pubescens* extract only had oviposition deterrence activity against *P. xylostella* female adults and did not affect mating. The caryophyllene oxide treatment group demonstrated repellent effects against both female and male adults of *P. xylostella* at 5 and 2.5 mg/mL; the oviposition deterrence results showed that the total egg numbers were decreased at 5 and 2.5 mg/mL, which indicated that caryophyllene oxide had repellent effects against both male and female adults, and also affected the mating rate and reduced the number of eggs laid. 

In our study, the behavioral responses of female and male adults of *P. xylostella* to 3,4-dimethoxycinnamic acid were not significantly different from those of the control. Previous studies have reported that linalool and γ-terpinene cause EAG responses, but neither male nor female adults could distinguish between control and these compounds [50]. The EAG responses to these compounds were not always consistent with the behavioral responses. In contrast, 3,4-dimethoxycinnamic acid showed strong oviposition deterrence activity, with an ODI value of 97.59% at 1.25 mg/mL. Phytophagous insects have two stages of oviposition behavior: flying direction, landing, drumming on the leaf surface, and laying eggs. The first stage is mediated by visual and olfactory cues; the second stage is mediated by gustatory cues [23,59]. Rhodojaponin-III showed oviposition deterrence activity against *Spodoptera litura* without depending on antenna responses [60]. These results suggest that the oviposition deterrence activity of 3,4-dimethoxycinnamic acid is unrelated to antennae olfactory behavior, and it may involve different sensory neurons. Whether 3,4-dimethoxycinnamic acid influences other parts of the olfactory or gustatory systems requires further research.

The emergence of resistance in insect pest species is the main problem that must be taken into account for a new management strategy. Typically, phytophagous insects treated with natural and synthetic insecticides can use detoxification enzymes to metabolize xenobiotics [31]. Previous studies have reported that the Car E and GST activities of *P. xylostella* larvae significantly decreased when in contact with caryophyllene oxide [31]. Thus, long-term caryophyllene oxide use may cause insecticide resistance. However, caryophyllene oxide exhibited repellent activity on *P. xylostella*, reduced the *P. xylostella* population, reduced the use of chemical pesticides, and delayed the development of insecticide resistance. In addition, it is not currently widely available on the market; no reports on caryophyllene oxide resistance in the field have been published.

This study confirmed that *A. pubescens* extract and its compound derivatives, caryophyllene oxide and 3,4-dimethoxycinnamic acid, have repellent oviposition effects on *P. xylostella*, and directly affect their interactions with cruciferous vegetables. In acute toxicity experiments with the oral administration of *A. pubescens* extract, the LD_50_ of mice was determined to be 7.35 ± 0.62 g/kg [61]; however, its use has not been reported in clinical applications [15]. The chemical caryophyllene oxide, a volatile of tomato flower [62] and tobacco [45], had a positive effect on two pollinators, *Apis mellifera* and *Bombus terrestris*, which are efficient and reliable pollinators of greenhouse crops, and a parasitic wasp of *Campoletis chlorideae*, which is the main parasitoid of *H. assulta* larvae. These studies indicated that caryophyllene oxide is relatively safe for non-target organisms.

The EAG and behavioral experiments demonstrated that the oviposition-repellent behavior of caryophyllene oxide on *P. xylostella* was related to the olfactory nervous system. Odor molecules bind to OR proteins upon ligand binding, leading to signaling via G-proteins and intracellular second messengers to open membrane ion channels [63]. This depolarizes the sensory neuron to generate action potentials, which are conducted along its axon into the olfactory bulb of the brain and induce behavioral responses to molecules, rather than acting on neurotransmitters related to insecticide resistance. Thus, *A. pubescens* extract, caryophyllene oxide, and 3,4-dimethoxycinnamic acid can delay the development of insecticide resistance.

In addition, our results from EAG and Y-tube olfactometer tests provide evidence that *P. xylostella* responds to volatile compounds from *A. pubescens*. Previous studies employing these methods have found that *Ocimum basilicum* oil and its constituents showed repellent activity against *Musca domestica* [26]. Cao et al. reported that *Panax notoginseng*, *Angelica sinensis*, *Gastrodia elata*, and *Peucedanum praeruptorum*, and their constituents, showed attractive activity against *Stegobium paniceum* [64]. Song et al. reported that *Pelargonium hortorum* oil and its constituents showed repellent oviposition against *P. xylostella* [4]. Thus, the EAG and Y-tube olfactometer tests can be used effectively in screening new potential repellent extracts.

Several studies have found that multi-component phytochemical blends have greater repellent activity against various insect species than the corresponding single semiochemicals [65]. For instance, the b-caryophyllene/hexanal blend, at a close natural ratio of 2:1, showed greater attraction to *Holotrichia parallela* than the compounds did individually [66]. Behavioral effects depending on different chemical contents in mixtures are critical for non-host avoidance, as observed for the eight volatiles in *Rosmarinus officinalis* at ratios of 13:2:13:8:1:24:6:17 having stronger repellent effects against *Ectropis obliqua* than the constituents alone in equal doses [50]. The five compounds of *Pelargonium hortorum* substances at ratios of 1:5:3:4:3 significantly repelled egg-laying of female DBM, considerably more than demonstrated by the independent compounds [4]. Therefore, in order to test the potential synergistic effect of the compounds found in extract, it is best to use the natural ratios of these compounds derived from plant extracts, which warrants further research.

## 5. Conclusions

Overall, this study demonstrated the oviposition deterrence and insecticidal activity potential of *A. pubescens* extract, caryophyllene oxide, and 3,4-dimethoxycinnamic acid against adult and larval stages of *P. xylostella*. Furthermore, the EAG and behavioral responses of *P. xylostella* adults to *A. pubescens* extract and caryophyllene oxide indicated that these compounds can be used to control moth populations by repelling both male and female adults of *P. xylostella*, thereby deterring oviposition in female adults. 3,4-dimethoxycinnamic acid demonstrated strong oviposition deterrence activity against female adults of *P. xylostella*, but it was not associated with the antennal olfactory responses. Our findings showed that *A. pubescens* extract and its constituents have potential applications in the development of natural insecticides and controlling moth populations. Further research is needed to determine the optimal proportions of these different components for insecticide applications and to examine the toxicological implications of these compounds. 

## Figures and Tables

**Figure 1 insects-14-00613-f001:**
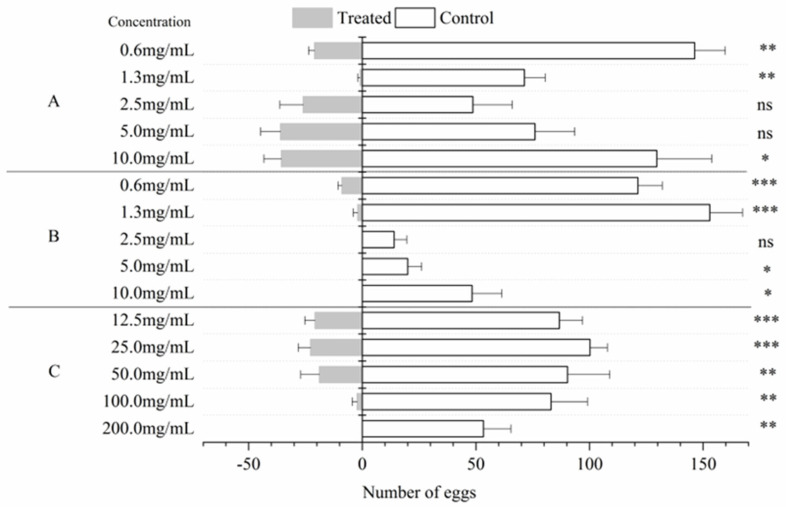
Number of *P. xylostella* eggs (mean ± SE) on leaves treated with five concentrations of 3,4-dimethoxycinnamic acid (**A**), caryophyllene oxide (**B**), *A. pubescens* extract (**C**), and with the control. Asterisks indicate significant differences: * = *p* < 0.05, ** = *p* < 0.01, *** = *p* < 0.001, ns = *p* > 0.05 (independent-samples *t*-test).

**Figure 2 insects-14-00613-f002:**
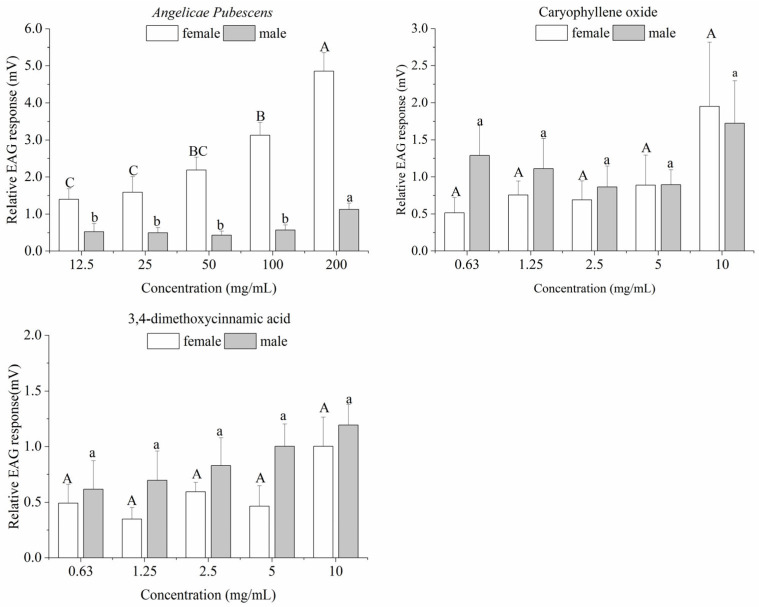
EAG responses (mean ± SE) of *P. xylostella* adults treated with five concentrations of *A. pubescens* extract, caryophyllene oxide, and 3,4-dimethoxycinnamic acid. Different letters represent significant differences at different concentrations (Tukey’s test, *p* < 0.05).

**Figure 3 insects-14-00613-f003:**
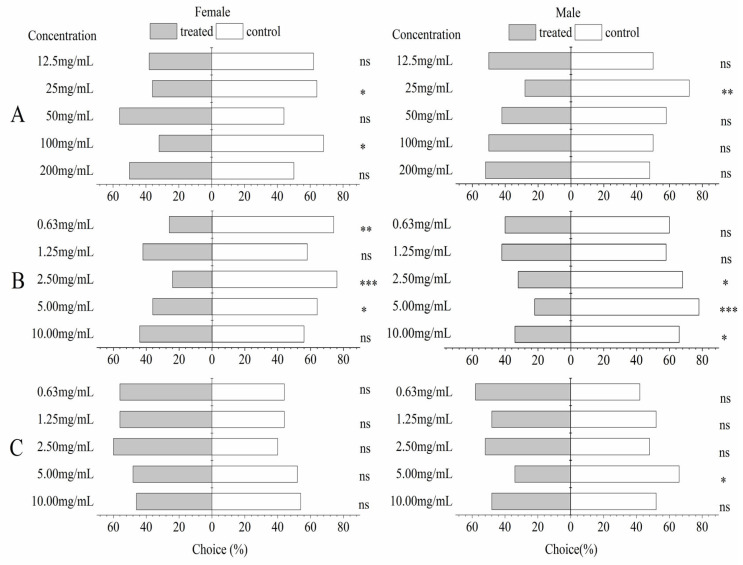
Behavioral responses of female and male *P. xylostella* adults to *A. pubescens* extract, caryophyllene oxide, and 3,4-dimethoxycinnamic acid in a Y-tube olfactometer. Both bars represent the percentages (percent choice n = 100) of moths that chose the treated (*A. pubescens* extract (**A**), caryophyllene oxide (**B**), and 3,4-dimethoxycinnamic acid (**C**) at five concentrations) or control arm. Asterisks indicate significant differences: * = *p* < 0.05, ** = *p* < 0.01, *** = *p* < 0.001, ns = *p* > 0.05 (Chi-squared (χ^2^) test).

**Table 1 insects-14-00613-t001:** Calibrated mortality of *P. xylostella* larvae to *A. pubescens* extract at five concentrations.

*A. pubescens* Extract Concentration	Calibrated Mortality (%)	Oviposition Deterrence Index (%)
200 mg/mL	100.00 ± 0.00 a	100.00 ± 0.00 a
100 mg/mL	97.00 ± 0.01 a	96.88 ± 0.03 ab
50 mg/mL	94.00 ± 0.04 a	69.80 ± 0.09 bc
25 mg/mL	40.00 ± 0.04 b	63.75 ± 0.08 c
12.5 mg/mL	33.00 ± 0.01 b	61.80 ± 0.08 c

Different letters represent significant differences at different concentrations (Tukey’s test, *p* < 0.05).

**Table 2 insects-14-00613-t002:** Chemical composition of 10 compounds (derived from the *A. pubescens* extract) used in subsequent tests against *P. xylostella*.

Sample	Retention Time	Composition (%)	Sample	Retention Time	Composition (%)
	(min)			(min)	
Artemisinin	6.337	0.163	Ethyl caffeate	9.284	0.079
4-Methylbenzylidene camphor	10.752	0.138	2-Hydroxycinnamic acid	1.207	0.074
Santonin	8.477	0.077	Caffeic acid	5.377	0.064
Caryophyllene oxide	10.959	0.072	3,4-Dimethoxycinnamic acid	11.527	0.014
Nootkatone	13.519	0.014	4-Coumaric acid	6.555	0.012

**Table 3 insects-14-00613-t003:** Calibrated mortality and oviposition deterrence index of 10 compounds from *A. pubescens* extract (at a concentration of 10 mg/mL) against *P. xylostella*.

Sample	Calibrated Mortality (%)	Oviposition Deterrence Index (%)
Artemisinin	30.00 ± 0.02 e	33.66 ± 0.03 bcd
4-Methylbenzylidene camphor	67.00 ± 0.06 bc	24.45 ± 0.04 cd
Santonin	61.00 ± 0.06 cd	48.46 ± 0.06 bc
Caryophyllene oxide	85.00 ± 0.05 ab	100 ± 0.00 a
Nootkatone	26.00 ± 0.08 e	100.00 ± 0.00 a
Ethyl caffeate	34.00 ± 0.03 e	32.08 ± 0.11 bcd
2-Hydroxycinnamic acid	33.00 ± 0.07 e	38.03 ± 0.06 bcd
Caffeic acid	40.00 ± 0.10 de	38.92 ± 0.01 bcd
3,4-Dimethoxycinnamic acid	75.00 ± 0.02 bc	57.01 ± 0.02 b
4-Coumaric acid	40.00 ± 0.02 de	90.67 ± 0.09 a
Matrine	100.00 ± 0.00 a	69.16 ± 0.02 b

Different letters represent significant differences at different concentrations (Tukey’s test, *p* < 0.05).

**Table 4 insects-14-00613-t004:** Lethal effects of *A. pubescens* extract, caryophyllene oxide, and 3,4-dimethoxycinnamic acid against third-instar larvae of *P. xylostella*.

Sample	Regression Equation	LC_50_ in mg/mL 95% CL	Chi-Square (DF = 23)	*p* Value
*A. Pubescens* extract	y = −3.69 + 2.78x	21.31(16.03−26.71)	71.74	0.00
Caryophyllene oxide	y = −1.16 + 1.76x	4.56(3.55−6.26)	46.86	0.02
3,4-Dimethoxycinnamic acid	y = −0.83 + 1.12x	5.52(4.00−8.87)	32.24	0.10
Matrine	y = −1.66 + 3.05x	3.50(2.99−4.13)	43.32	0.01

Each value represents the mean of five replicates, and each experiment included 20 individuals; 95%CL means confidence interval at the 95% confidence level.

## Data Availability

The authors confirm that all data are available in this paper.

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
