# Peer review of "Behavioral, Electrophysiological, and Toxicological Responses of Plutella xylostella to Extracts from Angelica pubescens"

_insects, 2023, doi:10.3390/insects14070613_

Round 1

Reviewer 1 Report

This article tackles with an issue of major concern that is to identify efficient and viable alternative to chemical pesticide to fight against insect pest. The article is quite easy to read and the methodology and results are clearly explained. However some important points are missing, particularly in the discussion section.

I would like the author to address the following point in order to improve the impact of this article :

Concerning the discussion section & global view of the paper :

A lot of paper are already published for the evaluation of the biological activity of plant extract and their components. The novelty of this paper is the exploration of the mode of action of this extract. In fact, as stated in the introduction, the authors wrote that they will start to explore the neuronal activity of the extract to start the determination of its mode of action. However, this point is not enough discussed in the paper, what should be improved. In particular, the authors could discuss about the effect for non-target species, as it is presented in the introduction section. As the stated purpose of the paper is to give informations for « integrated pest managment », it seems necessary to address some discussion point about the risk for human health, environment or at least side-effect on non-target insect species.

In the same line, the author underlight the emergence of resistance in insect pest species as main problem that must be take into account for new managment strategy. However, no test were done on resistant strains. To tackle with this issue, author could at least discussed the potential efficiency on resistant strain, according to the preliminary results that they obtained concerning extract mode of action.

It is important as the authors highlighted some neuronal effect of the extract and some of its components. Meaning that individuals which developped resistance to neurotoxic chemical pesticides could also be less sensitive to the plant extract. It is important to considere this point for the discussion of potential application in the field.

Concerning the methodology :

If possible, the author should precise the proportion of the different components in the pure extract. They tested the individual components at 1/10 compared to the complete extract. I wonder if it reflects their proportion in the extract.

What about testing the synergistic effect potentially involved in the biological efficency of the extract

Line 152 authors should precise the concentrations used for insecticidal assay

For Bioassay activity,the author should precise the criteria for the selection of the compounds and how the concentration of 10 mg/ml was selected.

Line 218 the authors present an ODI and mortality to larvae at 100 mg/ml but in the material and section section they wrote concentrations ranging from 0.63 to 10 mg/ml. Please clarifiy this point in the material and methods section. In the same way, the authors present a detterent effect of 100% when the extract is used at 200 mg/ml (Line 246 ). Is it consistent with the 96% of deterrent effect described for the concentration of 100 mg/ml ?

Concerning the results :

Line 277 The EAG responses of female moths treated with 3,4-dimethoxycinnamic acid at lower doses did not differ significantly, with values of 0.46, 0.59, 0.35, and 0.49 mv, respectively. » please precise significantly different from what ? and explain to which conditions correspond the different values presented

Line 338-339, please explain toward what caryophyllene revealed to be non-toxic ; in fact, if it could be a new candidate for new insecticides (as written) so it must be toxic (at least for pest species).

On the Figure 1, we observe that the component A has 100% of detterent effect for oviposition at 1.3 mg/ml but the biological effect is lower when the concentration increase. Can the author discussed this result, in line with the potential application as insecticide.

Some minor points to complete :

Line 78 it is not clear what the author mean with « single compounds could be better used to study functional mexhanisms and imrpoved the various effects »

Line 87 please correct the sentence « for to construct the for P. xylostella control »

Line 97-99 this sentence is not clear « .... which showed these oviposition deterrent responses are mainly driven by olfactory 98 system, are a possible mechanism for the development of repellents.... » ; would the author propose to test the use of the present protocol for the screening of new potential repellent extracts ?

Author Response

June 18, 2023

Response to Reviewer 1 Comments

Dear reviewer:

With this letter we resubmit our revised manuscript as required to insects (Manuscript No: insects-2381414). On behalf of all the contributing authors, I would like to express our sincere appreciations of reviewers' constructive comments and suggestions concerning our article entitled “Toxicity, electrophysiological and behavioural responses of the extract, pure compounds from Angelica pubescens against Plutella xylostella”. We have taken all suggestions into account, and hope that the responses will clarify the questions. The reviewer comments are laid out below in the black font and specific concerns have been numbered. Our response is given in the blue font and changes/additions to the manuscript are given in the red text and used “Track Changes” functions.

Reviewer #1

  1. A lot of paper are already published for the evaluation of the biological activity of plant extract and their components. The novelty of this paper is the exploration of the mode of action of this extract. In fact, as stated in the introduction, the authors wrote that they will start to explore the neuronal activity of the extract to start the determination of its mode of action. However, this point is not enough discussed in the paper, what should be improved. In particular, the authors could discuss about the effect for non-target species, as it is presented in the introduction section. As the stated purpose of the paper is to give informations for «integrated pest managment», it seems necessary to address some discussion point about the risk for human health, environment or at least side-effect on non-target insect species.

Response 1: We sincerely thank you for your valuable comments. We have checked the literature carefully and added more references into the Discussion part in the revised manuscript.

P12-13, line 454-461. Some previous studies reported that Nilaparvata lugens were repelled by caryophyllene oxide and found that NlugOBP8 highly expressed in multiple sensilla of N. lugens antenna. Antennae of dsNlugOBP-injected BPHs completely lost the ability to sense caryophyllene oxide [59,60]. Nguyen reported that on the basis of docking results, caryophyllene oxide with the mosquito odorant binding protein were ranked with their docking scores close permethrin, caryophyllene oxide exhibit larvicidal activity as potent mosquito odorant binding protein inhibitors [61]. Thus, we guessed that the caryophyllene oxide have repellent activity on P. xylostella may be function in the odorant binding protein of P. xylostella.

[59] He, P.; Chen, G.e.; Li, S.; Wang, J.; Ma, Y.; Pan, Y.; He, M. Evolution and functional analysis of odorant‐binding proteins in three rice planthoppers: Nilaparvata lugens, Sogatella furcifera, and Laodelphax striatellus. Pest Management Science 2019, 75.

[60] Duan, S.G.; Lv, C.L.; Liu, J.H.; Yi, S.C.; Yang, R.N.; Liu, A.; Wang, M.Q. NlugOBP8 in Nilaparvata lugens Involved in the Perception of Two Terpenoid Compounds from Rice Plant.

[61] Hung, N.H.; Quan, P.M.; Dai, D.N.; Satyal, P.; Huong, L.T.; Giang, L.D.; Hung, L.T.; Setzer, W.N. Environmentally-Friendly Pesticidal Activities of Callicarpa and Karomia Essential Oils from Vietnam and Their Microemulsions. Chemistry & biodiversity 2023, 20, e202200210, doi:10.1002/cbdv.202200210.

 P10, line 335-338. The use of solvent, with water 2-propanol and ethanol, sonication extraction for extracting compounds from Chinese medicinal plants. This method is attractive because it avoids the use of potentially harmful solvents, making it eco-friendlier and more easily solvent in water and potentially safer for using in the field.

 P13, line 511-520. This study confirmed that A. pubescens extract and its compounds caryophyllene oxide and 3, 4-dimethoxycinnamic acid as oviposition repellants, participate directly influenced in the interactions between P. xylostella and cruciferous vegetables. The extract of A. pubescens, were traditional Chinese medicine, through oral administration acute toxicity experiments, the LD50 of mice was determined to be 7.35 ± 0.62 g/kg [65], but have not been reported in clinical applications [15]. The chemical caryophyllene oxide, a volatile of tomato flower [66] and tobacco [46], had an attractive effect on two pollinators, Apis mellifera and Bombus terrestris, which are more efficient and reliable pollinators of greenhouse crops, and a parasitic wasp of Campoletis chlorideae, which the main endoparasitoid of H. assulta larvae. These literatures indicated that caryophyllene oxide was relatively safe for non-target organisms of bees.

[15] Lu, Y.; Wu, H.; Yu, X.; Zhang, X.; Wang, Z. Traditional Chinese Medicine of Angelicae Pubescentis Radix: A Review of Phytochemistry, Pharmacology and Pharmacokinetics. Frontiers in Pharmacology 2020, 11.

[46] Wang, C.; Li, G.; Miao, C.; Zhao, M.; Guo, X. Nonanal modulates oviposition preference in female Helicoverpa assulta (Lepidoptera: Noctuidae) via the activation of peripheral neurons. Pest Management Science 2020.

[65] Zhi-Wen, D.; Yu-Min, Z.; Hai-Shan, L.I.; Jia-Lun, W.; Ling, Z.; Wei, S. The Research on Acute and Long Time Toxicity of Angelicaroot Capsule. Journal of Shenyang Medical College 2002.

[66] Liu, J.; Zhang, J.; Shen, J.; Zhao, H.; Ma, W.; Jiang, Y. Differences in EAG Response and Behavioral Choices between Honey Bee and Bumble Bee to Tomato Flower Volatiles. Insects 2022, 13, doi:10.3390/insects13110987.

  1. In the same line, the author underlight the emergence of resistance in insect pest species as main problem that must be take into account for new managment strategy. However, no test were done on resistant strains. To tackle with this issue, author could at least discussed the potential efficiency on resistant strain, according to the preliminary results that they obtained concerning extract mode of action.

Response 2: We sincerely thank you for your valuable comments. We have checked the literature carefully and added more references into the Discussion part in the revised manuscript.

P13, line 489-497. The emergence of resistance in insect pest species as main problem that must be take into account for new management strategy. Generally, the phytophagous insects treated by natural and synthetic insecticides can use detoxification enzymes to metabolize xenobiotics [34]. Previous studies reported that the Car E and GST activities of P. xylostella larvae significantly decreased when contact to caryophyllene oxide [34]. Thus, long-term use of caryophyllene oxide may cause insecticides resistance. However, caryophyllene oxide have repellent activity on P. xylostella, reduced the population of P. xylostella, reduced the use of chemical pesticides and delaying the development of insecticides resistance and it has not been widely used in the market, and there is no literature on insecticides resistance at present.

[34] Huang, X.; Huang, Y.; Yang, C.; Liu, T.; Liu, X.; Yuan, H. Isolation and Insecticidal Activity of Essential Oil from Artemisia lavandulaefolia DC. against Plutella xylostella. Toxins 2021, 13, doi:10.3390/toxins13120842.

  1. It is important as the authors highlighted some neuronal effect of the extract and some of its components. Meaning that individuals which developped resistance to neurotoxic chemical pesticides could also be less sensitive to the plant extract. It is important to considere this point for the discussion of potential application in the field.

Response 3: We sincerely appreciate the valuable comments. We have checked the literature carefully and added more reference into the Discussion part in the revised manuscript.

P14, line 522-529. The EAG and behavioral experiments demonstrated that oviposition repellent behavior of caryophyllene oxide on P. xylostella related with the olfactory nervous system. The mechanism of olfactory was the odor molecules bind to OR proteins upon ligand binding, signal through G-proteins and intracellular second messengers to open membrane ion channels [67]. This depolarizes the sensory neuron to drive action potentials that are conduced along its axon into the olfactory bulb of the brain and responding behaviorally to molecules, rather than acting on neurotransmitters related to insecticides resistance. Thus, A. pubescens extract, caryophyllene oxide, and 3, 4-dimethoxycinnamic acid can delay the development of insecticides resistance.

[67] Sato, K.; Pellegrino, M.; Nakagawa, T.; Nakagawa, T.; Vosshall, L.B.; Touhara, K. Insect olfactory receptors are heteromeric ligand-gated ion channels. Nature 2008, 452, 1002-1006, doi:10.1038/nature06850.

  1. If possible, the author should precise the proportion of the different components in the pure extract. They tested the individual components at 1/10 compared to the complete extract. I wonder if it reflects their proportion in the extract. What about testing the synergistic effect potentially involved in the biological efficency of the extract. For Bioassay activity, the author should precise the criteria for the selection of the compounds and how the concentration of 10 mg/ml was selected.

Response 4:: We sincerely appreciate the valuable comments. We have checked the literature carefully, added some contents and we rewrote these sentences.

Sample

Retention time

Composition (%)

Sample

Retention time

Composition (%)

(min)

(min)

Artemisinin

6.337

0.163

Ethyl caffeate

9.284

0.079

4-Methylbenzylidene camphor

10.752

0.138

2-Hydroxycinnamic acid

1.207

0.074

Santonin

8.477

0.077

Caffeic acid

5.377

0.064

(-)-Caryophyllene oxide

10.959

0.072

3,4-Dimethoxycinnamic acid

11.527

0.014

Nootkatone

13.519

0.014

4-Coumaric acid

6.555

0.012

P6, line 243-244. Table 2 Chemical composition of 10 compounds from the A. pubescens extracts used in subsequent tests against P. xylostella

P 5, line 237-239. The polyene-alkynes and phenolic acids were main volatile constituents from A. pubescens extract. Thus, we selected ten compounds from the polyene-alkynes and phenolic acids from A. pubescens extract, with contents above 0.01%, for the following study.

P 14, line 538-549. Many studies have found that the multi-component blends of phytochemicals have a greater repellency against various insect species than the corresponding single semiochemicals [70]. For instance, the b-caryophyllene/hexanal blend at a close natural ratio of 2:1 showed attractive to Holotrichia parallela than individual compounds [71]. The behavioral effects depending upon difference chemicals content in mixtures are critical for non-host avoidance, as observed for the eight volatiles of 13:2:13:8:1:24:6:17 in R. officinalis repellent on E. obliqua was higher than the eight volatiles in the same dose and independent compounds [72]. The five compounds of 1: 5: 3: 4: 3 in Pelargonium hortorum substances significantly repelled the egg-laying of female DBM was higher than the independent compounds [4]. Therefore, in order to test the potential synergistic effect of the compounds from extract, it is best to use the natural ratio of these compounds from plant extracts, which needs further research in our experiments.

[4] Song, C.; Ma, L.; Zhao, J.; Xue, Z.; Yan, X.; Hao, C. Electrophysiological and Behavioral Responses of Plutella xylostella (Lepidoptera: Plutellidae) to Volatiles from a Non-host Plant, Geranium, Pelargonium hortorum (Geraniaceae). Journal of Agricultural and Food Chemistry 2022, 70.

[70] Hieu, T.T.; Choi, W.S.; Kim, S.I.; Wang, M.; Ahn, Y.J. Enhanced repellency of binary mixtures of Calophyllum inophyllum nut oil fatty acids or their esters and three terpenoids to Stomoxys calcitrans. Pest Management ence 2015, 71, 1213-1218.

[71] Zhang, M.; Cui, Z.; Zhang, N.; Xie, G.; Chen, L. Electrophysiological and Behavioral Responses of Holotrichia parallela to Volatiles from Peanut. Insects 2021, 12, 158.

[72] Zhang, Z.; Bian, L.; Sun, X.; Luo, Z.; Xin, Z.; Luo, F.; Chen, Z. Electrophysiological and behavioural responses of the tea geometrid Ectropis obliqua (Lepidoptera: Geometridae) to volatiles from a non-host plant, rosemary, Rosmarinus officinalis (Lamiaceae). Pest management science 2015, 71, 96-104, doi:10.1002/ps.3771

P 14, line 546-550. Our findings showed that A. pubescens extract and its constituents has potential applications in the development of insecticides and controlling moth populations. Further research is suggested to precise the proportion of the different components from the extract and explore the synergistic effect potentially involved in the biological efficiency of the extract, as well as to examine its effects on toxicological implications.

  1. Line 152 authors should precise the concentrations used for insecticidal assay. Line 218 the authors present an ODI and mortality to larvae at 100 mg/ml but in the material and section section they wrote concentrations ranging from 0.63 to 10 mg/ml. Please clarifiy this point in the material and methods section. In the same way, the authors present a detterent effect of 100% when the extract is used at 200 mg/ml (Line 246). Is it consistent with the 96% of deterrent effect described for the concentration of 100 mg/ml?

Response 5: As suggested by the reviewer, we have checked the manuscript carefully and added the test concentrations of plant extract and its compounds used for assays.

P 3, line 131-133. The 20% ethanol solvent with 0.5% Triton-X100 and 0.5% DMSO (TDE solution) acted as a solvent, the compound test solutions and matrine were configured as 10, 5, 2.5, 1.25, and 0.63 mg/mL.

P 4, line 145-146. A. pubescens extract was diluted in TDE solution to obtain 200, 100, 50, 25,12.5 mg/mL of test solution.

P 4, line 155-157. The A. pubescens extract (200, 100, 50, 25, 12.5 mg/mL) and its compounds of caryophyllene oxide and, 3,4-dimethoxycinnamic acid (10, 5, 2.5, 1.25, 0.63 mg/mL) as test solutions.

P 4, line 167-169. The A. pubescens extract (200, 100, 50, 25, 12.5 mg/mL) and its compounds of caryophyllene oxide and, 3,4-dimethoxycinnamic acid (10, 5, 2.5, 1.25, 0.63 mg/mL) as test solutions.

P 4, line 196-198. The behavioral response of P. xylostella adults to A. pubescens extract (200, 100, 50, 25, 12.5 mg/mL) and its constituents caryophyllene oxide and, 3,4-dimethoxycinnamic acid (10, 5, 2.5, 1.25, 0.63 mg/mL) were tested using a Y-tube olfactometer with an internal width of 24 mm × length of 20 cm (choice arm) and 29 mm × 20 cm (main arm) with a 30° angle between arms.

P 5, line 234-237. The result described in line 246 was consistent with that of 200mg/mL, and we selected 100mg/ml result to description. This sentence is not rigorous enough, which leads to ambiguity. We rewrote this sentence and added Table 1 in the manuscript. “In a previous study, of 20 plant species tested, the A. pubescens extract showed the oviposition deterrent and insecticidal activities against P. xylostella, with the oviposition deterrence index was 96.88% at 100 mg/mL, and the mortality to larvae was 97.00% at 100 mg/mL (Table 1).”

P 5-6, line 242-243.Table 1. Calibrated mortality of A. pubescens extract at five concentrations against larvae of P. xylostella.

A. pubescens extract Concentration

Calibrated mortality (%)

Oviposition deterrence index (%)

200 mg/mL

100.00±0.00a

100.00±0.00a

100 mg/mL

97.00±0.01a

96.88±0.03ab

50 mg/mL

94.00±0.04a

69.80±0.09bc

25 mg/mL

40.00±0.04b

63.75±0.08c

12.5 mg/mL

33.00±0.01b

61.80±0.08c

  1. Line 277 The EAG responses of female moths treated with 3,4-dimethoxycinnamic acid at lower doses did not differ significantly, with values of 0.46, 0.59, 0.35, and 0.49 mv, respectively. » please precise significantly different from what? and explain to which conditions correspond the different values presented.

Response 6: We sincerely appreciate the valuable comments. We have checked the literature carefully and we rewrote this sentence.

P 8, line 302-303. “The EAG responses of female moths treated by 3,4-dimethoxycinnamic acid did not differ significantly at 5, 2.5, 1.25, 0.63mg/mL, with values of 0.46, 0.59, 0.35, and 0.49 mv, respectively.”

  1. Line 338-339, please explain toward what caryophyllene revealed to be non-toxic; in fact, if it could be a new candidate for new insecticides (as written) so it must be toxic (at least for pest species).

Response 7: As suggested by the reviewer, we have added more reference to support this idea.

P 11, line 368-372. Importantly, Yulia [35] summarized the biological activity of β-caryophyllene and its oxide, they can easily penetrate cell membranes [36], do not cause allergic reactions upon contact, even in persons with dermatitis [37], increased the activity of cytochrome P450 in mice [38] and exhibited in vivo pain reveling and anti-inflammatory properties [39]. Caryophyllene revealed to be weak toxicity to human body at certain dose.

[35] Gyrdymova, Y.V.; Rubtsova, S.A. Caryophyllene and caryophyllene oxide: a variety of chemical transformations and biological activities. Chemical Papers 2022, 76, 1-39.

[36] Sarpietro, M.G.; Di Sotto, A.; Accolla, M.L.; Castelli, F. Interaction of β-caryophyllene and β-caryophyllene oxide with phospholipid bilayers: Differential scanning calorimetry study. Thermochimica Acta 2015, 600, 28-34.

[37] Matura, M.; Skld, M.; Brje, A.; Andersen, K.E.; Karlberg, A.T. Selected oxidized fragrance terpenes are common contact allergens. Contact Dermatitis 2010, 52, 320-328.

[38] Lněniková, K.; Svobodová, H.; Skálová, L.; Ambro, M.; Matouková, P. The Impact of Sesquiterpenes β-caryophyllene Oxide and trans nerolidol on Xenobiotic-metabolizing Enzymes in Mice in vivo. Xenobiotica 2017, 48, 1-29.

[39] Chavan, M.J.; Wakte, P.S.; Shinde, D.B. Analgesic and anti-inflammatory activity of Caryophyllene oxide from Annona squamosa L. bark. Phytomedicine: international journal of phytotherapy and phytopharmacology 2009, 17, 149-151.

  1. On the Figure 1, we observe that the component A has 100% of detterent effect for oviposition at 1.3 mg/ml but the biological effect is lower when the concentration increase. Can the author discussed this result, in line with the potential application as insecticide.

Response 8: We sincerely appreciate the valuable comments. We have checked the literature carefully and added more reference to support this idea.

P 11, line 385-398. In addition, we observed that 3,4-dimethoxycinnamic acid showed oviposition deterrence activity against P. xylostella with an ODI was 97.59% at 1.25 mg/mL, but the biological effect is lower when the concentration increase, with an ODI was 57.01% at 10 mg/mL. Previous reports about that the effect of the same volatile compound at different concentrations varies slightly. Such as the compound (Z)-9-octadecenoic acid ethyl ester (OE) is induced oviposition preference has a concentration-dependent bidirectional regulation: at high concentrations of OE inhibit oviposition while a low dose was attractive, with the Gr32a play a role in oviposition stimulant [44]. Linalool (10, 100 mg/L), and cis-3-hexen-1-ol (100 mg/L) exhibited repellent activities against Cyrtorhinus lividipennis, however, linalool (1 mg/L), and cis-3-hexen-1-ol (0.1 mg/L) exhibited attractive activities against C. lividipennis [45]. And the egg-laying preference of Helicoverpa assulta for heptanal disappeared when the concentration from 0.001 mol/L increased to 0.01 mol/L [46], Thus, 3,4-dimethoxycinnamic acid with the potential application as insecticide.

[44] Zhang, L.; Sun, H.; Grosse-Wilde, E.; Zhang, L.; Hansson, B.S.; Dweck, H.K.M. Cross-generation pheromonal communication drives Drosophila oviposition site choice. Curr Biol 2023, 33, 2095-2103.e2093.

[45] Jiang, N.; Mao, G. F,; Li, T.; Mo, J,C. 2018. The olfactory behavior response of Cytorhinus lividipennis to single component of rice volatiles [J]. Ecology and Environmental Sciences, 27(2): 262-267.

[46] Wang, C.; Li, G.; Miao, C.; Zhao, M.; Guo, X. Nonanal modulates oviposition preference in female Helicoverpa assulta (Lepidoptera: Noctuidae) via the activation of peripheral neurons. Pest management science 2020.

  1. Line 78 it is not clear what the author mean with «single compounds could be better used to study functional mexhanisms and imrpoved the various effects»

Response 9: We sincerely appreciate the valuable comments. We have checked the literature carefully and we deleted this sentence. P2, line 84.

  1. Line 87 please correct the sentence «for to construct the for P. xylostella control»;

Response 10: We sincerely appreciate the valuable comments. We have checked the literature carefully and we rewrote this sentence.

P 2, line 92-93. The “polyene-alkyne, and phenolic acid compounds could be adopted for to construct the for P. xylostella control” has been modified to “Thus, polyene-alkyne, and phenolic acid compounds could be used to control P. xylostella”.

  1. Line 97-99 this sentence is not clear « .... which showed these oviposition deterrent responses are mainly driven by olfactory system, are a possible mechanism for the development of repellents.... »; Would the author propose to test the use of the present protocol for the screening of new potential repellent extracts?

Response 11: We sincerely appreciate the valuable comments. We have checked the literature carefully and added more reference to support this idea.

P 3, line 102-107. The “The antennal and behavioural responses to the bioactive compounds, assessed using electroantennography and a Y-tube olfactometer, which showed these oviposition deterrent responses are mainly driven by olfactory system, are a possible mechanism for the development of repellents” has been modified to “The olfactory responses of P. xylostella to the bioactive compounds were investigated by electroantennographic tests to assess the sensitivity of females and males antennae to the test compounds and a Y-tube olfactometer bioassays to evaluate the insects' behavioral response of the same compounds, which will help to further elucidate the relationship between the oviposition deterrent activity of these compounds and olfactory behavior of P. xylostella. ”.

P 14, line 517-524. In addition, our results provide the evidence that P. xylostella responds to volatiles from A. pubescens by EAG and Y-tube olfactormeter tests. Previous studies also used these methods indicated that Ocimum basilicum oil and its constituents showed repellent response on Musca domestica [68]. Cao reported that the Panax notoginseng, Angelica sinensis, Gastrodia elata and Peucedanum praeruptorum and its constituents showed attractive activity on Stegobium paniceum [69]. Song reported that Pelargonium hortorum oil and its constituents showed oviposition repellants against P. xylostella [4]. Thus, the EAG and Y-tube olfactormeter tests were a proven protocol, which can be used screening of new potential repellent extracts.

[4] Song, C.; Ma, L.; Zhao, J.; Xue, Z.; Yan, X.; Hao, C. Electrophysiological and Behavioral Responses of Plutella xylostella (Lepidoptera: Plutellidae) to Volatiles from a Non-host Plant, Geranium, Pelargonium hortorum (Geraniaceae). Journal of Agricultural and Food Chemistry 2022, 70.

[68] Senthoorraja, R.; Subaharan, K.; Manjunath, S.; Pragadheesh, V.S.; Basavarajappa, S. Electrophysiological, Behavioural and Biochemical Effect of Ocimum Basilicum Oil and Its Constituents Methyl Chavicol and Linalool on Musca Domestica L. Environmental Science and Pollution Research, 1-14.

[69] Cao, Y.; Pistillo, O.M.; Lou, Y.; D'Isita, I.; Maggi, F.; Hu, Q.; Germinara, G.S.; Li, C. Electrophysiological and behavioural responses of Stegobium paniceum to volatile compounds from Chinese medicinal plant materials. Pest management science 2022, 78.

We tried our best to improve the manuscript and made some changes marked in red in revised paper which will not influence the content and framework of the paper. We appreciate for reviewers' warm work earnestly, and hope the correction will meet with approval. Once again, thank you very much for your comments and suggestions.

Thank you and best regards.

Sincerely yours,

Ruirui Zheng, Ph. D

Corresponding author: Chi Hao, Professor; Xizhong Yan, Associate Professor

College of Plant Protection, Shanxi Agricultural University, Taigu, 030800, China

E-mail addresses: sxauhc@163.com (C. Hao), yanxizhong80@163.com (X. Yan)

Reviewer 2 Report

Dear all,

After analyzing the manuscript, it was possible to raise some questions that need to be included:

Methodology:

Where were the insects collected to start the colony? What are the geographic coordinates? What are the characteristics of insects regarding the evolution of resistance? Have field insects been bred and used in experiments in the next generation? Or were they kept for how many generations?

These questions are essential for the evaluation of the paper, as it changes the entire justification.

There are many conflicts in the concepts used about the selection and management of resistance

Regards,

Author Response

June 18, 2023

Response to Reviewer 2 Comments

Dear reviewer:

With this letter we resubmit our revised manuscript as required to insects (Manuscript No: insects-2381414). On behalf of all the contributing authors, I would like to express our sincere appreciations of reviewers' constructive comments and suggestions concerning our article entitled “Toxicity, electrophysiological and behavioural responses of the extract, pure compounds from Angelica pubescens against Plutella xylostella”. We have taken all suggestions into account, and hope that the responses will clarify the questions. The reviewer comments are laid out below in the black font and specific concerns have been numbered. Our response is given in the blue font and changes/additions to the manuscript are given in the red text and used “Track Changes” functions.

Reviewer #2

Methodology:

  1. Where were the insects collected to start the colony? What are the geographic coordinates?

Response 1: P 3, line 122. As suggested by the reviewer, and we rewrote this sentence. “Plutella xylostella Larvae were collected from the vegetable fields of the Shanxi Agricultural University Experimental Station in Taigu County, Jinzhong City, Shanxi Province, China (37°25'22"N, 112°34'15"E), in April–May 2020 and were successive multi-generation reared in the laboratory of Shanxi Agricultural University. The rearing protocol was the same as that previously described by Yan [21] and Guo[22].”

[21] Yan, X.Z.; Deng, C.P.; Xie, J.X.; Wu, L.J.; Sun, X.J.; Hao, C. Distribution patterns and morphology of sensilla on the antennae of Plutella xylostella (L.)-A scanning and transmission electron microscopic study. Micron 2017, 103, 1-11.

[22] Zhao-Jiang., G.; Shi., K.; Qing-Jun., W.; You-Jun, Z. Technical specifications for the mass-rearing of the diamondback moth, Plutella xylostella (L.). Chinese Journal of Applied Entomology 2015, 52, 492-497.

  1. What are the characteristics of insects regarding the evolution of resistance?

Response 2: We sincerely appreciate the valuable comments. We have checked the literature carefully and we added the sentence.

P13, line 489-497. The emergence of resistance in insect pest species as main problem that must be take into account for new management strategy. Generally, the phytophagous insects treated by natural and synthetic insecticides can use detoxification enzymes to metabolize xenobiotics [34]. Previous studies reported that the Car E and GST activities of P. xylostella larvae significantly decreased when contact to caryophyllene oxide [34]. Thus, long-term use of caryophyllene oxide may cause insecticides resistance. However, caryophyllene oxide have repellent activity on P. xylostella, reduced the population of P. xylostella, reduced the use of chemical pesticides and delaying the development of insecticides resistance and it has not been widely used in the market, and there is no literature on insecticides resistance at present.

[34] Huang, X.; Huang, Y.; Yang, C.; Liu, T.; Liu, X.; Yuan, H. Isolation and Insecticidal Activity of Essential Oil from Artemisia lavandulaefolia DC. against Plutella xylostella. Toxins 2021, 13, doi:10.3390/toxins13120842.

  1. Have field insects been bred and used in experiments in the next generation? Or were they kept for how many generations?

Response 3:  We sincerely appreciate the valuable comments. We have checked the literature carefully and we added the sentence.

P 3, line 124.The F10 generation of P. xylostella was used for a bioassay.

  1. There are many conflicts in the concepts used about the selection and management of resistance

Response 4:  We sincerely appreciate the valuable comments. We have checked the literature carefully and rewrite these sentences in the revised manuscript.

P 1, line 23-25. Plant-based insecticides showed multi biological activity against pests due to their chemical constituents, and they have been considered promising insect control agents.  

P 2, line 58-61. Beyond their insecticidal properties, plant extracts have several advantages over traditional insecticides. For example, they are generally considered safe for human and animal use, biodegradable, and have lower toxicity for non-target organisms; these factors make them more environmentally acceptable than many synthetic insecticides.

P 10, line 328-329. Indiscriminate use of synthetic insecticides has caused a series of adverse effects to the environment and consumers and led to development of insecticide resistance of P. xylostella.

P 10, line 345-346. Previous studies have shown that plant extracts and derived bioactive compounds have a variety of biological activities to control pests, and they provides an alternative to synthetic insecticides due to these have biodegradable nature and safety to non-target organisms [27].

[27] Sengottayan; Senthil-Nathan. A Review of Resistance Mechanisms of Synthetic Insecticides and Botanicals, Phytochemicals, and Essential Oils as Alternative Larvicidal Agents Against Mosquitoes. Frontiers in physiology 2019, 10, 1591-1591.

We tried our best to improve the manuscript and made some changes marked in red in revised paper which will not influence the content and framework of the paper. We appreciate for editor/reviewers' warm work earnestly, and hope the correction will meet with approval. Once again, thank you very much for your comments and suggestions.

Thank you and best regards.

Sincerely yours,

Ruirui Zheng, Ph. D

Corresponding author: Chi Hao, Professor; Xizhong Yan, Associate Professor

College of Plant Protection, Shanxi Agricultural University, Taigu, 030800, China

E-mail addresses: sxauhc@163.com (C. Hao), yanxizhong80@163.com (X. Yan)

Reviewer 3 Report

Sheng et al. presented a very interesting study to identify two compounds from plant material against the global vegetable pest, Plutella xylostella. They extracted chemicals from Angelica pubescens and performed LC-MS to identify compounds. Then they conducted behavior assay, bioassay, EAG, etc to investigate the toxicity and physiological effects of these compounds on oviposition deterrent and olfaction repellency. The research is very well designed and data collection, analysis and presentation are scientifically sound.

Minor/prefer 

Title: “Behavioural” change to “behavioral”

Author Response

June 18, 2023

Response to Reviewer 3 Comments

Dear reviewer:

With this letter we resubmit our revised manuscript as required to insects (Manuscript No: insects-2381414). On behalf of all the contributing authors, I would like to express our sincere appreciations of reviewers' constructive comments and suggestions concerning our article entitled “Toxicity, electrophysiological and behavioural responses of the extract, pure compounds from Angelica pubescens against Plutella xylostella”. We have taken all suggestions into account, and hope that the responses will clarify the questions. The reviewer comments are laid out below in the black font and specific concerns have been numbered. Our response is given in the blue font and changes/additions to the manuscript are given in the red text and used “Track Changes” functions.

Reviewer #3

  1. Minor/prefer Title: “Behavioural” change to “behavioral”

Response 1:  Thanks for your careful checks. Based on your comments, we have made the corrections to make the word of “Behavioural” change to “behavioral” harmonized within the whole manuscript.

We tried our best to improve the manuscript and made some changes marked in red in revised paper which will not influence the content and framework of the paper. We appreciate for reviewers' warm work earnestly, and hope the correction will meet with approval. Once again, thank you very much for your comments and suggestions.

Thank you and best regards.

Sincerely yours,

Ruirui Zheng, Ph. D

Corresponding author: Chi Hao, Professor; Xizhong Yan, Associate Professor

College of Plant Protection, Shanxi Agricultural University, Taigu, 030800, China

E-mail addresses: sxauhc@163.com (C. Hao), yanxizhong80@163.com (X. Yan)

Round 2

Reviewer 2 Report

Dear,

The manuscript underwent improvements, but still has important questions that were not observed, such as: 1) What are the characteristics of insects regarding the evolution of resistance? The resistance ratio was not presented before the beginning of the experiments. It was not hot if there was a comparison with a population susceptible to referent. Anyway, the manuscript has important information regarding the evaluation of insects, but in relation to insect resistance, it needs to be revised, as it still has misconceptions and poorly presented.

Author Response

Response to Reviewer 2 Comments

Dear reviewer:

With this letter we resubmit our revised manuscript as required to insects (Manuscript No: insects-2381414). On behalf of all the contributing authors, I would like to express our sincere appreciations of reviewers' constructive comments and suggestions concerning our article entitled “Toxicity, electrophysiological and behavioural responses of the extract, pure compounds from Angelica pubescens against Plutella xylostella”. We have taken all suggestions into account, and hope that the responses will clarify the questions. The reviewer comments are laid out below in the black font and specific concerns have been numbered. Our response is given in the blue font and changes/additions to the manuscript are given in the red text and used “Track Changes” functions.

Reviewer #2

The manuscript underwent improvements, but still has important questions that were not observed, such as: 1) What are the characteristics of insects regarding the evolution of resistance? The resistance ratio was not presented before the beginning of the experiments. It was not hot if there was a comparison with a population susceptible to referent.

Response 1: We are sorry about the previous answer, somehow, we misunderstood the question. Now, I am answering the question as follows: In our study, the insects were collected from the vegetable fields of the Shanxi Agricultural University Experimental Station. For the eight years since 2016, the insects in this vegetable fields have been used for olfactory behavioral experiments and there no insecticides were applied during the cultivation. The insects were fed with fresh cabbage (the plants were grown from seeds in a plastic pot in a greenhouse on the Shanxi Agricultural University campus without any insecticides application) successive reared in the laboratory. As our laboratory has focused on research related to the olfactory behavioral of Plutella xylostella, thus, we did not measure the resistance ratio. Therefore, we do not know much about the characteristics of insects regarding the evolution of resistance. Your question is quite insightful and cutting-edge, and we will pay close attention to the research in the future.

  1. Song C, Ma L, Zhao J, Xue Z, Yan X, Hao C. Electrophysiological and Behavioral Responses of Plutella xylostella (Lepidoptera: Plutellidae) to Volatiles from a Non-host Plant, Geranium, Pelargonium × hortorum (Geraniaceae). J Agric Food Chem. 2022 May 25;70(20):5982-5992. doi: 10.1021/acs.jafc.1c08165.
  2. Wu, A, Li, X, Yan, X, Fan, W, Hao, C. Electroantennogram responses of Plutella xylostella (L.), to sex pheromone components and host plant volatile semiochemicals. J Appl Entomol. 2020; 144: 396– 406. doi:10.1111/jen.12744
  3. Yan XZ, Deng CP, Xie JX, Wu LJ, Sun XJ, Hao C. Distribution patterns and morphology of sensilla on the antennae of Plutella xylostella (L.)-A scanning and transmission electron microscopic study. Micron. 2017 Dec; 103:1-11. doi: 10.1016/j.micron.2017.08.002.
  4. Anyway, the manuscript has important information regarding the evaluation of insects, but in relation to insect resistance, it needs to be revised, as it still has misconceptions and poorly presented.

Response 2: We apologize for the confusion generated by the previous version of the manuscript and sincerely hope that our logic is now easier to follow with this new version.

P1, line 21-23. We deleted the sentence. The rapid development of P. xylostella insecticide resistance has been one of the main problems in their control; it has shown resistance to almost all insecticides traditionally used to control it.

P2, line 50-51. We rewrote the sentence. “Over the past few decades, excessive dependence on chemical pesticides to control P. xylostella has caused resistance to almost all traditional insecticides” were rewrote “Over the past few decades, excessive dependence on chemical pesticides to control P. xylostella, the P. xylostella has development resistance to traditional insecticides used to control it.

P10, line 343-344. We deleted the sentences. And lead to the development of insecticide resistance of P. xylostella. And we rewrote the sentence. Based on these environmental safety challenges, new pesticides to control P. xylostella need to be urgently developed.

Thank you and best regards.

Sincerely yours,

Ruirui Zheng, Ph. D

Corresponding author: Chi Hao, Professor; Xizhong Yan, Associate Professor

College of Plant Protection, Shanxi Agricultural University, Taigu, 030800, China

E-mail addresses: sxauhc@163.com (C. Hao), yanxizhong80@163.com (X. Yan)

Round 3

Reviewer 2 Report

Dear,

All sugestions werw analised.